# Fluoroquinolone and Second-Line Injectable Resistance Among Rifampicin- and Isoniazid-Resistant *Mycobacterium tuberculosis* Clinical Isolates: A Molecular Study from a High-Burden Setting

**DOI:** 10.3390/microorganisms13112470

**Published:** 2025-10-29

**Authors:** Rosângela Siqueira Oliveira, Angela Pires Brandao, Fabiane Maria de Almeida Ferreira, Sonia Maria da Costa, Vera Lucia Maria Silva, Lucilaine Ferrazoli, Erica Chimara, Juliana Maira Watanabe Pinhata

**Affiliations:** 1Núcleo de Tuberculose e Micobacterioses, Centro de Bacteriologia, Instituto Adolfo Lutz (IAL), São Paulo 01246-000, Brazil; rosangela.bio.pqc@gmail.com (R.S.O.); abrandao1502@gmail.com (A.P.B.); fabianeferreira2006@gmail.com (F.M.d.A.F.); lucilaine.ferrazoli@ial.sp.gov.br (L.F.); erica.chimara@ial.sp.gov.br (E.C.); 2Instituto Oswaldo Cruz, Fundação Oswaldo Cruz (FIOCRUZ), Rio de Janeiro 21040-360, Brazil

**Keywords:** gene sequencing, multidrug-resistant tuberculosis, line probe assay, second-line anti-tuberculosis drugs

## Abstract

Drug-resistant tuberculosis (DR-TB) threatens global TB control. We investigated the prevalence and molecular characteristics of second-line drug resistance among rifampicin (RIF)- and/or isoniazid (INH)-resistant *Mycobacterium tuberculosis* complex (MTBC) isolates in São Paulo, Brazil, using the MTBDR*sl* v. 2.0 line-probe assay. MTBC isolates RIF- and/or INH-resistant by GenoType MTBDR*plus* or phenotypic testing (2019–2021) were subsequently tested by MTBDR*sl* for fluoroquinolone (FQ) and injectable drugs (capreomycin, amikacin, kanamycin) resistance. Isolates with inferred mutations underwent Sanger sequencing. Of 13,557 isolates, 728 (5.4%) were RIF- and/or INH-resistant (297 INH-R, 235 RIF-R, 196 MDR). Among them, 623 (85.6%) were tested by MTBDR*sl*; 582 (93.4%) showed no additional resistance, while 41 (6.6%) carried mutations. FQ resistance was detected in 38 isolates (92.7%), mostly in *gyrA* (*n* = 35). Three isolates with *gyrB* mutations were wild-type by sequencing. Two MDR isolates harbored the *rrs* a1401g mutation, and one also harbored *gyrA* D94G. Sequencing confirmed resistance in 38 of 41 isolates. Most MDR strains with second-line mutations (*n* = 32/33; 97%) were pre-XDR. Affected patients were predominantly male (68.4%), with pulmonary TB (92.1%), and unfavorable outcomes (39.5%). Second-line resistance prevalence was low overall, but FQ resistance was high among MDR isolates. Findings support integrating molecular and sequencing-based tools for accurate detection and management of DR-TB.

## 1. Introduction

Drug-resistant tuberculosis (DR-TB) poses a significant challenge to global efforts aimed at reducing and eliminating the disease. In 2023, the World Health Organization (WHO) estimated 175,923 individuals were diagnosed with and treated for multidrug-resistant or rifampicin-resistant tuberculosis (MDR/RIF-R-TB). This accounts for only 44% of the estimated 400,000 MDR/RIF-R-TB global cases [1].

Pre-extensively drug-resistant TB (pre-XDR-TB) is caused by RIF-R/MDR *Mycobacterium tuberculosis* also resistant to a fluoroquinolone (FQ), while XDR-TB is a RIF-R/MDR strain resistant to a FQ and bedaquiline and/or linezolid [https://www.who.int/news/item/27-01-2021-who-announces-updated-definitions-of-extensively-drug-resistant-tuberculosis (accessed on 13 October 2025)]. The WHO recommends the 6-month all-oral BPaL and BPaLM regimens—bedaquiline, pretomanid, linezolid, and moxifloxacin—for RIF-R/MDR-TB susceptible to FQ. Furthermore, second-line injectable agents such as amikacin should be avoided due to their associated toxicity. The BPaL regimen was adopted by the Brazilian National Tuberculosis Control Program in 2024 [2].

The updated WHO definitions of pre-XDR and XDR-TB have influenced Brazil’s laboratory diagnostic algorithms. Routine testing currently relies on phenotypic drug susceptibility testing (pDST) for FQ and amikacin, or on Genotype MTBDR*sl* v. 2.0 (Bruker-Hain Lifescience, Nehren, Germany) line probe assays (LPAs). Under these revised definitions, such routine methods enable the detection of pre-XDR-TB only. In São Paulo, pDST for bedaquiline, clofazimine, delamanid, and linezolid was implemented at the end of 2024 to enable XDR-TB detection.

The LPA has been widely used for speeding up the detection of resistance to FQ and injectable aminoglycosides (amikacin and kanamycin) or cyclic peptides (capreomycin), supporting clinicians in the management of DR-TB treatment. The assay enables genetic identification of the *M. tuberculosis* complex (MTBC) and detection of resistance to FQ and injectables by targeting the most frequent mutations in the *gyrA*/*gyrB* and *rrs*/*eis* genes, respectively. It can be applied directly to clinical specimens or cultured isolates [3].

The Adolfo Lutz Institute serves as the public health reference laboratory for TB in the state of São Paulo, Brazil, which reported over 20,000 new TB cases in 2024. In 2019, Adolfo Lutz was the first public health reference laboratory in Brazil to implement GenoType MTBDR*plus* 2.0 (Bruker-Hain Lifescience) and MTBDR*sl* 2.0 LPAs for routine diagnosis of first- and second-line drug resistance, respectively. Molecular tests for drug resistance were incorporated into the routine diagnostic workflow in other Brazilian regions at a later stage. Because of the relatively late introduction of molecular testing and the absence of national drug resistance surveys, knowledge of DR-TB molecular basis remains limited.

Very few studies have evaluated the use of the MTBDR*sl* 2.0 assay on clinical isolates in high-throughput laboratories. In Brazil, available data on mutations associated with second-line drug resistance remain scarce. The studies conducted to date relied on smaller sample sizes and were performed before the implementation of the current definitions of pre-XDR and XDR-TB, as well as before the introduction of new TB treatment regimens [4,5]. To address this gap, this comprehensive real-world study aimed to describe the mutations associated with resistance to FQ and second-line injectable drugs in MTBC isolates resistant to RIF and/or isoniazid (INH), and to assess their prevalence among all eligible isolates received at the public health reference laboratory of the state of São Paulo between 2019 and 2021. This study focuses on isolates resistant to RIF and/or INH because second-line drugs are specifically indicated for the treatment of DR-TB.

We hypothesized that molecular characterization of RIF-R and/or INH-resistant (INH-R) MTBC isolates in São Paulo can detect prevalent mutations associated with FQ and second-line injectable resistance, providing essential data to support regional surveillance and guide treatment strategies.

## 2. Materials and Methods

### 2.1. Clinical Isolates and Study Population

Mycobacterial cultures are routinely submitted from regional laboratories within the TB network of São Paulo. All MTBC isolates that tested RIF-R and/or INH-R by GenoType MTBDR*plus* 2.0 assay between January 2019 and December 2021, and that were subsequently tested by MTBDR*sl*, were included in the study. Additionally, isolates identified as RIF-R and/or INH-R by pDST using the BACTEC MGIT 960 system (Becton Dickinson, Sparks, MD, USA) at other laboratories from the network and referred to our laboratory for second-line testing by MTBDR*sl* during the study period, were also included. Additional isolates from the same patient were considered only when resistance results differed from those of previous isolates.

Patients’ data were obtained from TBWeb, an online notification system for TB cases in the state of São Paulo. A new TB case was defined as a patient who had never been treated or had received treatment for less than 30 days. Relapse cases included patients who had been previously treated and declared cured. Retreatment cases referred to patients who had received treatment for more than 30 days but had interrupted it for 30 consecutive days, or those who started a new treatment regimen following drug resistance detection.

### 2.2. GenoType MTBDRplus and MTBDRsl v. 2.0 Line-Probe Assays (LPA)

DNA extraction and PCR were performed for both assays according to the manufacturer’s instructions [6,7]. For samples in which RIF-R and/or INH-R was detected, the same DNA extract used for MTBDR*plus* was also used for MTBDR*sl*. The resulting PCR products were hybridized with probes immobilized on strips using the GT-Blot 48 automated system (Bruker-Hain Lifescience). Results of both assays were analyzed and interpreted by a trained professional according to WHO recommendations [8], using GenoScan software v. 3.4.134 (Bruker-Hain Lifescience).

An inferred mutation was defined by the absence of hybridization with both the wild-type (WT) and corresponding mutant (MUT) probe(s). Inconclusive results by the LPAs were retested, and only isolates with truly inferred mutations underwent Sanger sequencing.

### 2.3. Gene Sequencing

Sanger sequencing was used to characterize inferred mutations in the MTBDR*plus* and MTBDR*sl* targets, as next-generation sequencing (NGS) was not implemented in routine diagnostics at our laboratory during the study period. This approach reliably confirmed the mutations detected by the LPAs.

For the *rpoB* gene, the pair of primers amplified a 350 bp fragment covering the RRDR (positions 1184–1533 from the start codon) [9]. The *katG* was sequenced from positions −135 upstream of *katG* start codon to +431 nucleotides downstream from the gene’s end by using five pairs of primers [10]. The primers for *inhA* promoter amplified 248 bp fragment at positions −168 to +80 from the start codon of *fabG1* gene [11]. The *gyrA* and *gyrB* were sequenced at nucleotide positions 155–475 and 1213–1625, respectively, while the *rrs* gene was sequenced from position 881 to 23 nucleotides downstream of its 3′ end [5].

The reaction mixtures (25 µL) contained 1× buffer, 1.75 mM MgCl2 for *rpoB* (1.65 mM for *rrs*, 1.5 mM for *katG*/*inhA*/*gyrA*/*gyrB*), 0.75 U Taq polymerase (GoTaq G2 Flexi, Promega, Madison, WI, USA), 200 µM each dNTP (dNTP mix, Promega), 10 pmol primers for *rpoB*/*gyrA*/*gyrB*/*rrs* (5 pmol for *katG*/*inhA*), approximately 50 ng of DNA template, and nuclease-free water (Qiagen, Germantown, MD, USA). Amplification included initial denaturation at 94 °C for 5 min, followed by 40 cycles: denaturation at 94 °C for 1 min, annealing at 60 °C for *rpoB* (62 °C for *katG*/*gyrA*/*gyrB*, 64 °C for *inhA,* 66 °C for *rrs*) for 1 min, extension at 72 °C for 1 min, and a final extension at 72 °C for 10 min. Primers and unincorporated nucleotides were enzymatically removed (ExoSAP-it, Affymetrix, Santa Clara, CA, USA) and the amplicons were sequenced using BigDye Terminator v3.1 (Applied Biosystems, Foster City, CA, USA). The sequence data generated on the ABI 3130 × L Genetic Analyzer (Applied Biosystems) were analyzed with BioEdit v7.2.5, BLAST [http://www.ncbi.nlm.nih.gov/BLAST (accessed on 2 October 2025)], and BioEdit v. 7.2.5 tools [12].

Discordance was defined as any discrepancy between MTBDR*sl* results and gene sequencing in the detection of mutations associated with drug resistance. Discordant results were analyzed qualitatively, with each case reviewed to determine the likely cause of discrepancy, such as heteroresistance or limitations of the assay.

## 3. Results

### 3.1. Clinical Isolates

Between January 2019 and December 2021, a total of 13,557 isolates subjected to MTBDR*plus* or pDST to RIF and INH were identified as MTBC. Of these, 12,829 (94.6%) showed no resistance-associated mutations or were susceptible by pDST, 297 (2.2%) were INH-R, 235 (1.7%) RIF-R, and 196 (1.4%) MDR. Of the 297 INH-R isolates, 284 (95.6%) were tested using the MTBDR*sl* assay. Among the RIF-R and MDR isolates, 154 of 235 (65.5%) and 185 of 196 (94.4%), respectively, were also tested by MTBDR*sl*, resulting in a total of 623 (85.6%) isolates screened for FQ and second-line injectables resistance of the total 728 isolates with any first-line resistance.

At the initial stage of MTBDR*sl* implementation in 2019, isolates presenting inferred or inconclusive results in molecular assays were referred for pDST using the MGIT 960 system; therefore, not all, but the majority of INH/RIF or MDR isolates were tested by MTBDR*sl* in our study.

Of the 623 isolates tested with MTBDR*sl*, 551 (88.3%) were tested by MTBDR*plus* at our laboratory, while 72 (11.7%) were tested by MGIT 960 at other laboratories (Figure 1).

### 3.2. MTBDRsl Results

Of the 623 isolates subjected to MTBDR*sl*, 582 (93.4%) showed no mutations in the target genes, while 41 (6.6%) harbored mutations (Table 1). Among the 284 INH-R tested, 281 (98.9%) did not show mutations. Similarly, 96.8% (*n* = 149/154) of RIF-R and 82.2% (*n* = 152/185) of MDR isolates had no mutations by MTBDR*sl*.

As shown in Table 1, among the 582 isolates without mutations by MTBDR*sl*, nearly half (48.3%, *n* = 281) were INH-R, while 26.1% (*n* = 152) were MDR and 25.6% (*n* = 149) were RIF-R. Of the 41 isolates with mutations, 38 (95.0%) showed FQ resistance-associated mutations—35 in *gyrA*, 2 in *gyrB*, and 1 in both genes.

Among the remaining three isolates with mutations, two isolates (one RIF-R and one MDR) harbored *rrs* mutations conferring resistance to second-line injectables, and another MDR isolate showed mutations in both *gyrA* and *rrs* (Table 1).

### 3.3. Mutations Detected by Gene Sequencing

Table 2 presents the mutations detected by MTBDR*plus*, MTBDR*sl*, and/or gene sequencing in the 41 isolates with second-line resistance-associated mutations by MTBDR*sl*. The *gyrA* D94G mutation was the most prevalent, identified in 13 of the 41 isolates (31.7%) isolates. It was detected as a single mutation in 10 isolates (including two exhibiting heteroresistance), co-occurring with *gyrA* D94N/Y in two isolates, and in association with *rrs* a1401g in one isolate (Table 2, Figure 2).

The *gyrA* S91P mutation was detected in 11 of the 41 isolates (26.8%), all of which were MDR. This was followed by the *gyrA* A90V mutation, found in seven isolates (17.1%), also exclusively among MDR isolates. Only one isolate showed an inferred mutation in *gyrA* by MTBDR*sl*, characterized by a WT1-negative result. Sequencing revealed the presence of the G88C mutation, which is associated with FQ resistance (Table 2, Figure 2).

All three isolates presenting an inferred *gyrB* WT-negative mutation on MTBDR*sl*—two RIF-R with only this mutation and one INH-R also showing the *gyrA* D94N/Y mutation—were wild-type confirmed by sequencing. Thus, the two isolates with *gyrB* mutations alone, initially classified as FQ-resistant by MTBDR*sl*, were reclassified as susceptible after sequencing (Table 2).

Regarding *rrs*, in addition to the isolate with *gyrA* and *rrs* mutations by MTBDR*sl*, another isolate showed an *rrs* mutation—inferred WT1-negative—which characterized this isolate as resistant to second-line injectables. However, sequencing revealed this was a false resistance result (since inferred mutations in LPA should be interpreted as resistant), identifying this isolate as wild-type (Table 2). Therefore, among the 41 isolates initially classified as resistant by MTBDR*sl*, 38 were truly resistant (36 to FQ, one to injectables, and one to both), as sequencing classified two isolates with *gyrB* inferred mutations only and one isolate with *rrs* inferred mutation only as wild-type.

Overall, 32 of 33 MDR isolates showed additional resistance to FQ on MTBDR*sl*; therefore, these isolates were classified as pre-XDR, including one with the *rrs* a1401g mutation also. Therefore, 32 of the 38 isolates (84.2%) presenting second-line resistance mutations were pre-XDR.

### 3.4. Clinical Characterization of the Patients

Each of the 38 isolates presenting resistance to second-line drugs corresponded to a distinct patient. Table 3 shows the characteristics of these patients. The median age range was 36.5 years, and 68.4% (*n* = 26) were male. Most patients (92.1%, *n* = 35) had pulmonary TB, and 86.8% (*n* = 33) were HIV-negative. New TB cases accounted for 52.6% (*n* = 20) of the cohort. Regarding treatment outcomes, 60.5% (*n* = 23) of the patients were cured, while 23.7% died (*n* = 9), 13.2% (*n* = 5) were lost to follow-up, and 2.6% (*n* = 1) had treatment failure.

## 4. Discussion

This study provides an overview of the prevalence and patterns of molecular resistance to second-line drugs among MTBC isolates in São Paulo, Brazil, over a three-year period.

Among the 13,557 MTBC isolates subjected to MTBDR*plus* or pDST, 728 (5.4%) showed resistance to RIF and/or INH. Regarding first-line drug resistance in our setting, INH-R was the most prevalent (2.2%), followed by RIF-R (1.7%) and MDR (1.4%). Although Brazil is classified as a high TB burden country, it is not among those with the highest prevalence of DR-TB, with an estimated 2000 cases of DR-TB annually [1].

The lower proportion of RIF-R isolates compared to INH-R and MDR isolates (65.5%, 95.6%, and 94.4%, respectively) tested by MTBDR*sl* in our study is due to the particular high prevalence of borderline *rpoB* mutations in our region, which appear as inferred on LPA [13]. Brandão et al. [13] reported a high proportion (55.1%) of RIF-discordant *M. tuberculosis* isolates in São Paulo—resistant by Xpert MTB/RIF but susceptible by MGIT 960—most of which carried borderline *rpoB* mutations, predominantly H445N. These isolates were largely concentrated within actively circulating clusters, suggesting a significant role of specific mutations in local transmission dynamics.

In this study, from January to December 2019, all isolates with inferred or inconclusive mutations in any of the genes targeted by MTBDR*plus* were submitted to pDST. At that time, only pDST results were considered due to limited knowledge of the clinical significance of inferred mutations. As most of these borderline mutations often result in low-level increases in RIF minimum inhibitory concentrations (MICs), many of such isolates tested RIF-susceptible by MGIT. Thus, they were not referred for MTBDR*sl* testing.

Although not all isolates harboring mutations conferring resistance to first-line drugs were tested with MTBDR*sl*, the vast majority (85.6%) underwent second-line drug testing. This extensive coverage minimizes potential bias and supports the representativeness of the analyzed sample. Nonetheless, we acknowledge that incomplete testing could have introduced a minor degree of bias in the estimated prevalence of second-line resistance, particularly among RIF-R isolates, which were tested with MTBDR*sl* at a lower frequency than INH-R and MDR isolates. This limitation should therefore be taken into account when interpreting the findings.

Among the 623 isolates tested, 585 (93.9%) showed no mutations, indicating that second-line resistance remains relatively infrequent in São Paulo. The low prevalence of second-line drug resistance observed in this study has important public health implications. In Brazil, the recommended regimen for MDR-/RIF-R-TB is the all-oral BPaL regimen; however, individualized treatment regimens are still used in specific cases, in which FQ may be included based on DST results and clinical evaluation [2]. The low frequency of FQ resistance observed supports their continued use as effective components in such individualized regimens.

Of the 38 isolates (6.1%) with mutations detected by MTBDR*sl* and confirmed by sequencing, 37 (97.4%) carried FQ resistance-associated mutations in *gyrA*, excluding two isolates with *gyrB* mutations by LPA that were identified as wild type by sequencing. This supports previous findings that quinolone resistance-determining region (QRDR) mutations in *gyrA* are the main cause of FQ resistance in *M. tuberculosis* [14,15].

Regarding *gyrA*, the most frequently detected mutation in our study was D94G, followed by S91P and A90V, consistent with previous reports from São Paulo [4,5]. For instance, Matsui et al. [4] analyzed 156 MDR isolates for resistance to FQ and injectable drugs and also identified D94G as the predominant *gyrA* mutation, followed by D94N, A90V, and S91P. Similarly, Pinhata et al. [5] analyzed 365 MDR isolates and identified D94G and A90V as the most common *gyrA* mutations.

In other Latin American countries, previous studies from Colombia and Peru have reported FQ resistance rates of approximately 13% among MDR-TB patients [16,17], comparable to the rate observed in our study (17.8%).

Variants such as D94G, D94H, D94N, and D94Y are consistently associated with high-level FQ resistance, whereas A90V, S91P, and D94A generally confer low to intermediate resistance levels [14,15]. The D94G substitution, in particular, has been linked to elevated MICs and poorer clinical outcomes, including increased odds of treatment failure or death among RIF-R-/MDR-TB patients. In contrast, S91P and A90V may retain partial susceptibility to MFX at higher concentrations, although their clinical impact appears less pronounced than that of D94G [18,19,20,21,22,23]. The G88C mutation, detected in the only isolate with an inferred *gyrA* mutation by MTBDR*sl*, has also been associated with high-level FQ resistance [24] and was previously reported in São Paulo [5].

Interestingly, three isolates showing *gyrB* WT-negative results and one isolate with *rrs* WT1-negative on MTBDR*sl*, were confirmed to be wild-type by sequencing, revealing potential false resistance results by the LPA. This finding emphasizes the importance of sequencing or pDST for drug resistance confirmation when MTBDR*sl* results suggest inferred mutations. In line with our observation, Seifert et al. [25] reported that approximately 15% of isolates with only WT probe loss by MTBDR*plus* or MTBDR*sl* were wild-type by sequencing, confirming that this pattern does not always indicate true resistance.

Notably, among the 185 MDR isolates tested with MTBDR*sl*, 152 (82.2%) did not show mutations. However, 32 out of 33 MDR isolates with second-line resistance were also FQ-resistant, thus meeting the definition of pre-XDR. This highlights the concerning emergence of pre-XDR TB, a major challenge due to limited treatment options [26].

FQ resistance among MDR-TB patients in our study was lower than reported elsewhere. For instance, a study from Pakistan documented a notably high rate of FQ resistance among MDR-TB patients (52.7%) [27], while in South Korea resistance to FQ was identified in 26.2% of MDR-TB cases [28]. Similarly, data from India revealed a FQ resistance rate of 27.3% among MDR-TB isolates [29], and whole-genome sequencing data from England demonstrated FQ resistance in 23.9% of MDR-TB patients [30]. These findings highlight global FQ resistance spread and the urgent need for expanded drug testing and access to newer TB treatments.

The demographic profile of the 38 patients harboring isolates with second-line resistance mutations revealed a predominance of males (68.4%), with a median age of 36.5 years. Most patients (92.1%) had pulmonary TB and were HIV-negative (86.8%). More than half of the patients (52.6%) were new TB cases, indicating ongoing primary DR-TB transmission in São Paulo. The rate of unfavorable outcomes among DR-TB patients was high (39.5%). Nevertheless, it is not possible to directly attribute these outcomes to drug resistance, as they are also affected by other factors—including HIV status, comorbidities, and prior TB treatment—that were not controlled for in this analysis.

### Study Limitations

This study has limitations that should be considered when interpreting the findings. First, not all isolates with resistance to first-line drugs were tested using the MTBDR*sl* assay, which may have resulted in underestimation of the true prevalence of FQ or second-line injectable drug resistance. Another limitation is that pDST for second-line drugs was not systematically performed, limiting direct comparison with genotypic findings and potentially affecting the confirmation of resistance.

As the study relied mainly on molecular assays, some degree of misclassification cannot be ruled out, particularly in cases involving uncommon or uncharacterized mutations. Lastly, sequencing was performed only for isolates with inferred mutations rather than for all samples, which may have led to undetected resistance mutations not covered by MTBDR*sl*. This approach was adopted primarily due to the greater complexity and turnaround time associated with sequencing. Given these constraints, sequencing was reserved for cases requiring confirmation or clarification of genotypic results, as its routine application remains less feasible in high-throughput diagnostic settings.

## 5. Conclusions

The low overall prevalence of second-line resistance among MTBC isolates in São Paulo is encouraging; however, the elevated frequency of FQ resistance among MDR isolates with second-line mutations highlights the need for routine additional drug resistance testing in all MDR-TB cases. Furthermore, false resistance cases inferred by the LPA underscore the critical need for confirmatory testing—such as sequencing or, alternatively, pDST—to ensure accurate resistance detection and strengthen molecular surveillance efforts.

Future directions include expanding the use of NGS to enhance the detection of resistance-associated mutations and integrating molecular data into Brazil’s national TB surveillance network to strengthen monitoring and guide evidence-based treatment policies.

## Figures and Tables

**Figure 1 microorganisms-13-02470-f001:**
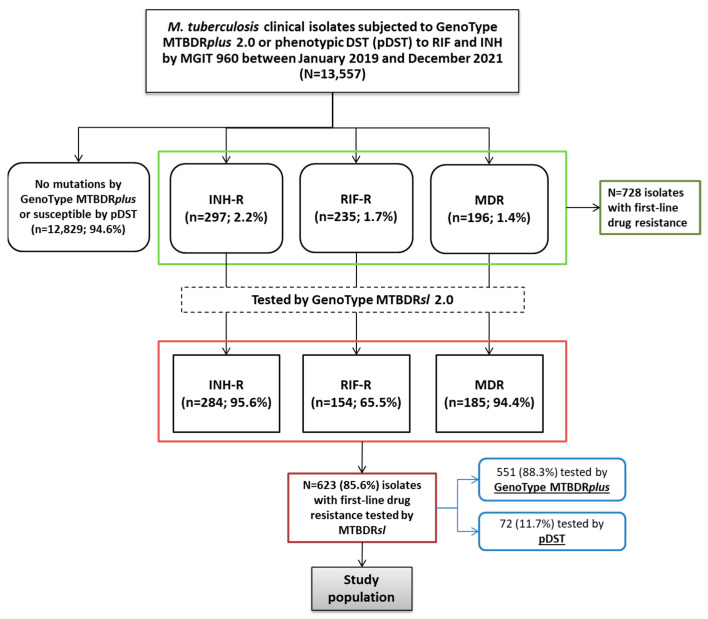
Flowchart of *Mycobacterium tuberculosis* isolates included in the study. The diagram shows the distribution of isolates based on initial resistance screening by GenoType MTBDR*plus* or phenotypic DST (pDST), followed by testing with the GenoType MTBDR*sl* 2.0 assay. Numbers and percentages indicate the proportion of isoniazid-resistant (INH-R), rifampicin-resistant (RIF-R), and multidrug-resistant (MDR) isolates at each step.

**Figure 2 microorganisms-13-02470-f002:**
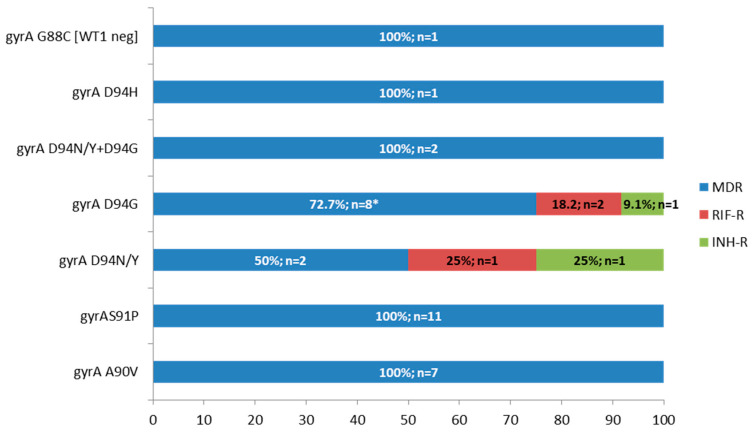
Frequency and number of *M. tuberculosis* isolates with *gyrA* mutations according to drug resistance type, MDR: multidrug-resistant, RIF-R: rifampicin-resistant, INH-R: isoniazid-resistant, G88C mutation was detected by gene sequencing and the MTBDR*sl* result is in brackets (WT1 probe negative, i.e., inferred mutation), * one isolate with D94G had also *rrs* a1401g, and two were heteroresistant.

**Table 1 microorganisms-13-02470-t001:** MTBDR*sl* results of the 623 *Mycobacterium tuberculosis* isolates with resistance to rifampicin and/or isoniazid.

	MTBDR*sl* Results						
	No. (%) Isolates with No Mutations	No. (%) Isolates with Mutations FQ Resistance			No. (%) Isolates with Mutations Injectables Resistance	No. (%) Isolates with Mutations FQ + Injectables Resistance	
**First-Line Drug Resistance**	**Wild-Type**	* **gyrA** *	* **gyrB** *	* **gyrA + gyrB** *	* **rrs** *	* **gyrA + rrs** *	**Total**
**Isoniazid**	281 (48.28)	1 (2.8)	1 (50)	1 (100)	0	0	284 (45.6)
**Rifampicin**	149 (25.60)	3 (8.6)	1 (50)	0	1 (50)	0	154 (24.7)
**MDR**	152 (26.12)	31 (88.6)	0	0	1 (50)	1 (100)	185 (29.7)
**Total**	582 (100)	35 (100)	2 (100)	1 (100)	2 (100)	1 (100)	623 (100)

FQ: fluoroquinolones, MDR: multidrug-resistant (i.e., resistant to rifampicin and isoniazid).

**Table 2 microorganisms-13-02470-t002:** Mutations detected by MTBDR*plus,* MTBDR*sl*, and/or gene sequencing in the isolates tested by MTBDR*sl*.

Mutations Detected by MTBDR*sl*/Sequencing	MTBDR*plus* or MGIT Result	Mutations Detected by MTBDR*plus*/Sequencing	No. Isolates	Frequency (%)
*gyrA*				
mut1 [A90V] ^a^ *N* = 7	MDR	*rpoB* H445Y/*inhA* C-15T	1	2.9
		*rpoB* S450L/*katG* S315T1	5	14.3
		Not performed (MGIT result) ^#^	1	2.9
mut2 [S91P] ^a^ *N* = 11	MDR	*rpoB* H445Y/*katG* S315T1/*inhA* C-15T	1	2.9
		*rpoB* S450L/*katG* S315T1	1	2.9
		*rpoB* S450L/*inhA* C-15T	9	25.7
mut3b [D94N/Y] ^b^ *N* = 3	RIF-R	*rpoB* H445N (1333 c > a)	1	2.9
	MDR	*rpoB* H445D/*katG* S315T1	2	5.7
mut3c [D94G] ^b^ *N* = 8	RIF-R	*rpoB* S450L	1	2.9
		*rpoB* H445N (1333 c > a)	1	2.9
	MDR	*rpoB* D435V/*katG* S315T1	1	2.9
		*rpoB* M434I (1302 g > a) + H445N (1333 c > a)/*katG* S315T1/*inhA* C-15T	1	2.9
		*rpoB* H445C (1333 ca > tg)/*katG* S315T1	1	2.9
		*rpoB* S450L/*katG* S315N (944 g > a) + A379T (1135 g > a)/*inhA* C-15T	3	8.6
wt3 + mut3c [WT + D94G] ^b^ *N* = 2	INH-R	inhA C-15T	1	2.9
	MDR	*rpoB* S450L/*katG* S315T1	1	2.9
mut3b + mut3c [D94N/Y + D94G] ^b^ *N* = 2	MDR	*rpoB* WT8 + S450L/*katG* WT + S315T1	1	2.9
		*rpoB* S450L/*katG* S315N (944 g > a) + A379T (1135 g > a)/*inhA* C-15T	1	2.9
mut3d [D94H] ^b^	MDR	*rpoB* S450L/*katG* S315T1	1	2.9
wt1 absent [G88C by sequencing] *^a^	MDR	*rpoB* S450L/*katG* S315T1	1	2.9
Total *gyrA*	-	-	35	100.0
** *gyrB* **				
wt absent ^a^ [WT by sequencing] * *N* = 2	RIF-R	*rpoB* L452P (1355 t > c)	1	50.0
	INH-R	* inhA * C-15T	1	50.0
** *gyrA + gyrB* **				
mut3b [D94N/Y] ^b^ + wt neg ^a^ [WT by sequencing] *	INH-R	Not performed (MGIT result) ^#^	1	100.0
***rrs* mutation**				
mut1 [a1401g]	MDR	*rpoB* WT7 + H445Y/*katG* S315T1	1	50.0
wt1 absent [WT by sequencing] *	RIF-R	rpoB H445N	1	50.0
** *gyrA + rrs* **				
mut3c [D94G ] ^b^ + mut1 [a1401g]	MDR	*rpoB* D435V/*katG* S315T1	1	100.0

Underlined are mutations for which MTBDR*plus* has specific probes, * isolates with inferred mutations by MTBDR*sl* and sequenced, sequencing results are in [ ], mut: mutation probes included in the MTBDR*sl* assay, WT: wild-type (i.e., isolates without mutations), ^#^ isolates not submitted to MTBDR*plus* and that underwent pDST by MGIT 960, ^a^ mutation associated with low-level resistance to fluoroquinolones, ^b^ mutation associated with high-level resistance to fluoroquinolones, INH-R: isoniazid-resistant, MDR: multidrug-resistant (i.e., resistant to rifampicin and isoniazid), RIF-R: rifampicin-resistant.

**Table 3 microorganisms-13-02470-t003:** Characteristics of the 38 patients with TB resistant to fluoroquinolones and/or amikacin.

Characteristics	No. Total (%) *n* = 38		No. MDR (%) *n* = 33	
**Age ***	36.5 ± 12 (range 19–63)		34 ± 11 (range 19–63)	
**Sex**				
Male	26	(68.4)	21	(63.6)
Female	12	(31.6)	12	(36.4)
**Clinical presentation**				
Pulmonary	35	(92.1)	30	(90.1)
Pulmonary and extrapulmonary	2	(5.3)	2	(6.1)
Extrapulmonary	1	(2.6)	1	(3.0)
**HIV status**				
Negative	33	(86.8)	28	(84.8)
Positive	5	(13.2)	5	(15.2)
**Type of TB case**				
New	20	(52.6)	16	(48.5)
Retreatment	12	(31.6)	12	(36.4)
Relapse	6	(15.8)	5	(15.2)
**FQ/injectables resistance (gene mutated)**				
FQ (*gyrA*)	36	(94.7)	31 ^#^	(93.9)
Injectables (*rrs*)	1	(2.6)	1	(3.0)
FQ + injectables (*gyrA + rrs*)	1	(2.6)	1 ^#^	(3.0)
**Outcome**				
Cure/Treatment completed	23	(60.5)	20	(60.6)
Death from TB	6	(15.8)	5	(15.2)
Death from other causes	3	(7.9)	3	(9.1)
Loss of follow up	5	(13.2)	4	(12.1)
Failure	1	(2.6)	1	(3.0)

* Age values expressed as mean ± standard deviation, ^#^ classified as pre-XDR (pre-extensively drug-resistant), FQ: fluoroquinolones, MDR: multidrug-resistant (i.e., resistant to rifampicin and isoniazid).

## Data Availability

The original data presented in the study are openly available in GenBank at https://www.ncbi.nlm.nih.gov/ under the accession number PRJNA888434 (accessed on 1 October 2025).

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
