# Peer review of "Fluoroquinolone and Second-Line Injectable Resistance Among Rifampicin- and Isoniazid-Resistant Mycobacterium tuberculosis Clinical Isolates: A Molecular Study from a High-Burden Setting"

_microorganisms, 2025, doi:10.3390/microorganisms13112470_

Round 1

Reviewer 1 Report

Comments and Suggestions for Authors

This work is relevant due to the global problem posed by tuberculosis and the limited information available about mutations associated with resistance to second-line antibiotics in Brazil, but it doesn’t emphasize the magnitude of the local problem (São Paulo as a “high-burden setting”).

It is suggested to make the following improvements:

  • Although it is mentioned that Brazil is a country with a high TB burden, the text does not contextualize how these findings compare with the situation in other high-burden countries. Including a comparative reference would strengthen the connection to the global landscape. The resistance frequencies to FQs or injectables are not directly compared with international data, which would enrich the analysis.
  • Strengthen the transition to the reason why the study focuses on resistant isolates to RIF and INH (MDR or potential precursors to XDR).
  • The description of mutations could be organized by distinguishing between the most frequent and less common ones, associating each with high or low levels of resistance. This would improve the interpretive robustness of the findings.
  • The results section is quite lengthy and could be subdivided into paragraphs, for example, a general description of the isolate cohort, results of the MTBDRsl, mutations detected and confirmed by sequencing, and clinical characterization of the patients.
  • Repetition is observed in expressions such as “showed no mutations” or “did not harbor mutations.”
  • There are some style inconsistencies, such as alternating use of “WT-negative” and “wild-type,” which could be unified (for example, “wild type confirmed by sequencing”). Phrases like “the other – also MDR – showed mutations” could be rewritten as “another MDR isolate showed mutations” to maintain a more formal tone.
  • Several long sentences could be divided into shorter segments.
  • There is repetition of expressions such as “in our setting” or “second-line resistance.”

Author Response

Reviewer 1

Comments and Suggestions for Authors

This work is relevant due to the global problem posed by tuberculosis and the limited information available about mutations associated with resistance to second-line antibiotics in Brazil, but it doesn’t emphasize the magnitude of the local problem (São Paulo as a “high-burden setting”).

Response: We thank the reviewer for recognizing the relevance of this study and the importance of addressing gaps in knowledge about mutations associated with second-line drug resistance in Brazil.

It is suggested to make the following improvements:

  • Although it is mentioned that Brazil is a country with a high TB burden, the text does not contextualize how these findings compare with the situation in other high-burden countries. Including a comparative reference would strengthen the connection to the global landscape. The resistance frequencies to FQs or injectables are not directly compared with international data, which would enrich the analysis.

Response: We appreciate this valuable suggestion. We have now included a brief comparison with data from other high-burden countries to provide better global context (thirteenth paragraph of the Discussion section). This addition strengthens the international relevance of our findings.

  • Strengthen the transition to the reason why the study focuses on resistant isolates to RIF and INH (MDR or potential precursors to XDR).

Response: We appreciate the reviewer’s suggestion. The manuscript has been revised to improve the transition and clarify the rationale for focusing on isolates resistant to RIF and INH, as can be found in the end of the Introduction.

  • The description of mutations could be organized by distinguishing between the most frequent and less common ones, associating each with high or low levels of resistance. This would improve the interpretive robustness of the findings.

Response: We thank the reviewer for this thoughtful suggestion. However, we preferred to maintain the original organization of the table – according to the mutations detected by MTBDRsl probes. Nevertheless, information regarding the levels of resistance to fluoroquinolones (high or low) has now been included in the legend of Table 2 to enhance the interpretive value of the data.

  • The results section is quite lengthy and could be subdivided into paragraphs, for example, a general description of the isolate cohort, results of the MTBDRsl, mutations detected and confirmed by sequencing, and clinical characterization of the patients.

Response: The Results section was subdivided according to the reviewer’s suggestions.

  • Repetition is observed in expressions such as “showed no mutations” or “did not harbor mutations.”

Response: The whole text was revised to avoid these repetitions.

  • There are some style inconsistencies, such as alternating use of “WT-negative” and “wild-type,” which could be unified (for example, “wild type confirmed by sequencing”). Phrases like “the other – also MDR – showed mutations” could be rewritten as “another MDR isolate showed mutations” to maintain a more formal tone.

Response: The suggestions were accepted and integrated in the manuscript.

  • Several long sentences could be divided into shorter segments.

Response: Long sentences were revised and shortened.

  • There is repetition of expressions such as “in our setting” or “second-line resistance.”

Response: The whole text was revised to avoid these repetitions.

Reviewer 2 Report

Comments and Suggestions for Authors

The manuscript's strengths

  1. Relevant Research Question: The study addresses drug-resistant tuberculosis (DR-TB), a global health priority, with a focus on fluoroquinolone and injectable resistance in a high-burden setting.

  2. Large Dataset: The inclusion of >13,000 isolates, with over 600 tested for second-line resistance, adds weight to the findings.

  3. Use of Molecular and Sequencing Approaches: Combining MTBDRsl assays with Sanger sequencing strengthens diagnostic accuracy and enhances the reliability of resistance detection.

  4. Clear Clinical Relevance: The observation of high fluoroquinolone resistance among MDR isolates provides important evidence for TB control programs and treatment strategies.

  5. Well-structured Results: Data are systematically presented with clear tables and flowcharts that improve readability.Major Comments

  1. Incomplete Resistance Testing: Not all isolates with first-line resistance were tested with MTBDRsl. This could bias prevalence estimates of second-line resistance. Authors should discuss the representativeness of the tested sample more explicitly.

  2. Limited pDST: Phenotypic confirmation was not systematically performed for second-line drugs. Without pDST, the study relies heavily on molecular assays, which may misclassify resistance in some cases. This limitation should be emphasized more clearly, especially regarding clinical interpretation.

  3. Restricted Sequencing Application: Sequencing was only used for inferred mutations rather than all isolates. This leaves potential undetected resistance mutations not covered by MTBDRsl probes. Expanding sequencing coverage, or discussing why it was limited, would strengthen the work.

  4. Potential Overinterpretation of Clinical Impact: The manuscript links mutation patterns with treatment outcomes (cure, death, failure) but does not control for other confounders (HIV status, comorbidities, prior treatment). Interpretation should be toned down or adjusted.

  5. Clarity on Definitions: Terms such as “false resistance” and “pre-XDR” should be clarified with reference to WHO definitions. For example, in cases where sequencing reclassified isolates as wild-type, emphasize that molecular assays alone may not be definitive.

  6. Generalizability: Findings are based on isolates from São Paulo, Brazil, a specific epidemiological setting. The discussion should address whether results can be extrapolated to other high-burden countries.

Minor Comments

  1. Abstract: The sentence “Most resistant strains (78%) were pre-XDR” could be misleading since the overall prevalence of second-line resistance was low. Suggest revising to highlight that the proportion was high within MDR isolates.

  2. Grammar/Stylistics: Some sentences are lengthy and could be made more concise, e.g., in the introduction (lines 34–49).

  3. Figures/Tables: Ensure that abbreviations (INH-R, RIF-R, MDR, pre-XDR) are consistently defined in figure legends and tables for readability.

  4. References: Some references (e.g., WHO guidelines, prior Brazilian studies) should be cited more specifically when comparing prevalence rates.

  5. Ethics Statement: While informed consent was waived, authors may consider explicitly clarifying that patient identifiers were not accessed or disclosed.

  6. Typographical Issues: A few minor typographical inconsistencies exist (e.g., spacing around percentage values, “FQs resistance” should be “FQ resistance”).

Comments on the Quality of English Language

The manuscript is generally well written and understandable, with appropriate scientific terminology. However, there are several areas where the English could be improved for clarity and conciseness:

  • Some sentences, particularly in the Introduction and Discussion, are long and complex, which reduces readability. Shorter, more direct sentences would improve flow.
  • Minor grammatical issues are present (e.g., “FQs resistance” should be “FQ resistance,” spacing around percentages, subject–verb agreement in some instances).

  • Inconsistent use of abbreviations (e.g., RIF-R vs. RIF resistant) should be standardized.

  • Occasional awkward phrasing (“presented an inferred mutation,” “false-resistance result”) could be smoothed for clarity.

Author Response

Reviewer 2

The manuscript's strengths

  1. Relevant Research Question: The study addresses drug-resistant tuberculosis (DR-TB), a global health priority, with a focus on fluoroquinolone and injectable resistance in a high-burden setting.
  2. Large Dataset: The inclusion of >13,000 isolates, with over 600 tested for second-line resistance, adds weight to the findings.
  3. Use of Molecular and Sequencing Approaches: Combining MTBDRsl assays with Sanger sequencing strengthens diagnostic accuracy and enhances the reliability of resistance detection.
  4. Clear Clinical Relevance: The observation of high fluoroquinolone resistance among MDR isolates provides important evidence for TB control programs and treatment strategies.
  5. Well-structured Results: Data are systematically presented with clear tables and flowcharts that improve readability.

Response: We sincerely thank the reviewer for these positive and encouraging comments. We appreciate the recognition of the study’s relevance, methodological robustness, and clarity of data presentation. The feedback reinforces the value of our approach and findings in contributing to the understanding of drug-resistant tuberculosis in high-burden settings.

Major Comments

  1. Incomplete Resistance Testing: Not all isolates with first-line resistance were tested with MTBDRsl. This could bias prevalence estimates of second-line resistance. Authors should discuss the representativeness of the tested sample more explicitly.

Response: We appreciate the reviewer’s observation. Although not all isolates harboring first-line drug resistance were tested with the MTBDRsl assay, the vast majority (85.6%) were included in second-line resistance testing. Therefore, we believe this does not significantly affect the representativeness of the sample. Nonetheless, we have revised the text to clarify this point, acknowledging that incomplete testing could introduce minor bias in the estimated prevalence of second-line resistance (fifth paragraph of the Discussion).

  1. Limited pDST: Phenotypic confirmation was not systematically performed for second-line drugs. Without pDST, the study relies heavily on molecular assays, which may misclassify resistance in some cases. This limitation should be emphasized more clearly, especially regarding clinical interpretation.

Response: We appreciate the reviewer’s insightful comment. We acknowledge that phenotypic drug susceptibility testing for second-line drugs was not systematically performed, and that reliance on molecular assays alone may lead to occasional misclassification of resistance. We have now emphasized this limitation more clearly in the Discussion, noting that molecular results should be interpreted with caution, particularly in the absence of phenotypic confirmation, and that discrepancies between genotypic and phenotypic results may have clinical implications.

  1. Restricted Sequencing Application: Sequencing was only used for inferred mutations rather than all isolates. This leaves potential undetected resistance mutations not covered by MTBDRsl probes. Expanding sequencing coverage, or discussing why it was limited, would strengthen the work.

Response: We thank the reviewer for this valuable comment. Sequencing was performed only for isolates with inferred to confirm the genotypic results, rather than for all isolates. We acknowledge that this approach may have limited the detection of resistance mutations not covered by MTBDRsl probes. We have now clarified this methodological limitation in the Discussion.

  1. Potential Overinterpretation of Clinical Impact: The manuscript links mutation patterns with treatment outcomes (cure, death, failure) but does not control for other confounders (HIV status, comorbidities, prior treatment). Interpretation should be toned down or adjusted.

Response: We thank the reviewer for this important observation. We agree that treatment outcomes may be influenced by additional clinical and epidemiological factors such as HIV status, comorbidities, and history of prior treatment, which were not controlled for in this analysis. We have therefore revised the text to acknowledge this limitation and have toned down the interpretation of associations between mutation patterns and treatment outcomes. We now emphasize that these findings should be interpreted with caution and considered hypothesis-generating rather than conclusive (fourteenth paragraph of Discussion).

  1. Clarity on Definitions: Terms such as “false resistance” and “pre-XDR” should be clarified with reference to WHO definitions. For example, in cases where sequencing reclassified isolates as wild-type, emphasize that molecular assays alone may not be definitive.

Response: We thank the reviewer for this important comment. The term “pre-XDR” has already been defined in the Introduction in accordance with WHO 2021 criteria (MDR-TB with additional resistance to any fluoroquinolone). Regarding “false resistance,” we emphasize that in cases where sequencing reclassified isolates as wild-type, molecular assays alone may not provide definitive results. We have clarified in the text that such discrepancies highlight the importance of confirming inferred resistance mutations with sequencing or phenotypic drug susceptibility testing (end of Discussion).

  1. Generalizability: Findings are based on isolates from São Paulo, Brazil, a specific epidemiological setting. The discussion should address whether results can be extrapolated to other high-burden countries.

Response: We thank the reviewer for this observation. We have complemented the Discussion by including data from other high-burden countries, highlighting similarities and differences in fluoroquinolone and second-line drug resistance patterns (thirteenth paragraph of the Discussion).

Minor Comments

  1. Abstract: The sentence “Most resistant strains (78%) were pre-XDR” could be misleading since the overall prevalence of second-line resistance was low. Suggest revising to highlight that the proportion was high within MDR isolates.

Response: We thank the reviewer for this suggestion. The sentence has been revised to clarify that most MDR strains with second-line mutations (n=32/33; 97%) were pre-XDR, thus referring specifically to MDR isolates, rather than the overall sample.

  1. Grammar/Stylistics: Some sentences are lengthy and could be made more concise, e.g., in the introduction (lines 34–49).

Response: We thank the reviewer for this comment. The manuscript has been carefully revised to shorten lengthy sentences, including those in the Introduction, to improve clarity and readability.

  1. Figures/Tables: Ensure that abbreviations (INH-R, RIF-R, MDR, pre-XDR) are consistently defined in figure legends and tables for readability.

Response: We thank the reviewer for this suggestion. All abbreviations (INH-R, RIF-R, MDR, pre-XDR) are now consistently defined in figure legends and tables to enhance readability.

  1. References: Some references (e.g., WHO guidelines, prior Brazilian studies) should be cited more specifically when comparing prevalence rates.

Response: We thank the reviewer for this suggestion. We have revised the manuscript to discuss more specifically prevalence rates of drug resistance in the Discussion.

  1. Ethics Statement: While informed consent was waived, authors may consider explicitly clarifying that patient identifiers were not accessed or disclosed.

Response: We thank the reviewer for this comment. We have revised the Ethics Statement to clarify that, although informed consent was waived, no patient identifiers were accessed or disclosed during the study, ensuring confidentiality and compliance with ethical standards.

  1. Typographical Issues: A few minor typographical inconsistencies exist (e.g., spacing around percentage values, “FQs resistance” should be “FQ resistance”).

Response: We thank the reviewer for noting these typographical issues. All minor inconsistencies have been corrected throughout the manuscript, including spacing around percentage values and changing “FQs resistance” to “FQ resistance.”

Comments on the Quality of English Language

The manuscript is generally well written and understandable, with appropriate scientific terminology. However, there are several areas where the English could be improved for clarity and conciseness:

  • Some sentences, particularly in the Introduction and Discussion, are long and complex, which reduces readability. Shorter, more direct sentences would improve flow.

Response: Long sentences along the whole manuscript were revised and shortened.

  • Minor grammatical issues are present (e.g., “FQs resistance” should be “FQ resistance,” spacing around percentages, subject–verb agreement in some instances).

Response: Grammatical issues were resolved.

  • Inconsistent use of abbreviations (e.g., RIF-R vs. RIF resistant) should be standardized.

Response: RIF-R abbreviation was standardized.

  • Occasional awkward phrasing (“presented an inferred mutation,” “false-resistance result”) could be smoothed for clarity.

Response: Awkward phrasing was corrected throughout the manuscript.

Reviewer 3 Report

Comments and Suggestions for Authors

The manuscript by Oliveira et. al explore the mutation associated with fluoroquinolones and SLID among the RIF/INH or MDR TB strains. The data reinforces the threat of MDR TB and XDR TB, further constant surveillance is required to optimize the therapy for such patients. Overall, the manuscript looked fine and needed a few modifications before being accepted. Please see the comments below

  1. Line 39: 400000 is total number of TB cases not the MDR.RIF-R TB cases, correction needed.
  2. Line 125-128: Just to allow reader to follow add a line that these number of isolates were screened for fluoroquinolone and SLID resistance.
  3. Is there any certain reason not all INH/RIF or MDR strains were not analyzed by MTBDRsl 2.0.
  4. I would suggest to add the following in the legends of table 2,  mut 1, mut2  and other mut and also describe what is WT so that is table can be self-explanatory.
  5. The two gyrB isolates and one isolate with rrs mutation should be highlighted or marked in table itself as susceptible on sequencing.
  6. M. tuberculosis should italicized throughout the text.

Author Response

Reviewer 3

Comments and Suggestions for Authors

The manuscript by Oliveira et. al explore the mutation associated with fluoroquinolones and SLID among the RIF/INH or MDR TB strains. The data reinforces the threat of MDR TB and XDR TB, further constant surveillance is required to optimize the therapy for such patients. Overall, the manuscript looked fine and needed a few modifications before being accepted. Please see the comments below

Response: We thank the reviewer for the positive evaluation and for highlighting the importance of continuous surveillance of drug-resistant tuberculosis. We agree that ongoing monitoring of resistance patterns is essential to guide effective treatment strategies and public health interventions.

  1. Line 39: 400000 is total number of TB cases not the MDR.RIF-R TB cases, correction needed.

Response: We thank the reviewer for this comment. However, according to the Global Tuberculosis Report 2024 from the World Health Organization, 400,000 represent the total number of MDR/RIF-R TB cases worldwide, while the total number of TB cases is 10.8 million. This information is available at https://www.who.int/teams/global-programme-on-tuberculosis-and-lung-health/tb-reports/global-tuberculosis-report-2024. For this reason, no changes were made to the sentence.

  1. Line 125-128: Just to allow reader to follow add a line that these number of isolates were screened for fluoroquinolone and SLID resistance.

Response: This information was added in the first paragraph of the Results section.

  1. Is there any certain reason not all INH/RIF or MDR strains were not analyzed by MTBDRsl 2.0.

Response: We appreciate the reviewer’s question. At the beginning of the implementation of the MTBDRsl 2.0 test in 2019, isolates showing inferred or inconclusive results in molecular assays were referred for phenotypic drug susceptibility testing using the MGIT 960 system. For this reason, not all — but the majority — of INH/RIF or MDR isolates were analyzed by MTBDRsl. This information was included in the second paragraph of the Results section.

Over time, as knowledge regarding the interpretation of resistance categories evolved and recommendations were issued to allow final reporting based solely on molecular test results, the procedure changed and some isolates were no longer analyzed by phenotypic DST. One example is the WHO expert group recommendation in 2021 that any mutation in the RRDR of the rpoB gene (except silent ones) should be assumed to confer RIF resistance, thereby eliminating the need for phenotypic DST (WHO Technical report on critical concentrations for drug susceptibility testing of isoniazid and the rifamycins (rifampicin, rifabutin and rifapentine). CC BY-NC-SA 3.0 IGO. Geneva, 2021. page 84).

  1. I would suggest to add the following in the legends of table 2,  mut 1, mut2  and other mut and also describe what is WT so that is table can be self-explanatory.

Response: The suggested changes were included in Table 2 legend.

  1. The two gyrB isolates and one isolate with rrs mutation should be highlighted or marked in table itself as susceptible on sequencing.

Response: Table 2 was revised and the information ‘WT by sequencing’ was included for the isolates without mutations by sequencing.

  1. M. tuberculosis should italicized throughout the text.

Response: Correction was made.

Reviewer 4 Report

Comments and Suggestions for Authors

 Introduction

  • The introduction provides adequate global and national context, but it should clearly articulate the knowledge gap, why São Paulo’s molecular resistance data remain underexplored and how this study adds to regional surveillance literature.
  • The WHO definitions of pre-XDR and XDR-TB are included, but the transition from global guidance to local relevance is abrupt. Strengthen by briefly connecting how these evolving definitions affect diagnostic policies in Brazil.
  • Citations [4]–[5] and [10] overlap conceptually with this study. You could emphasize what distinguishes your dataset (e.g., post-2021 MTBDRsl roll-out) to highlight originality.
  • Add a concise statement of hypothesis or primary objective at the end of the introduction (e.g., “We hypothesized that…”) to orient readers toward the analytic focus.

Methods

  • The methods are well structured but require clearer justification for inclusion/exclusion of isolates. For example, specify the rationale for excluding duplicate isolates from the same patient and describe how discordant results were handled statistically or qualitatively.
  • The sample size (n = 13,557) is impressive, yet the selection pathway (from initial isolates to analysed subset) should be summarized in a short paragraph before Figure 1 for transparency.
  • The gene targets and sequencing protocols are described in technical detail, but the section should have:
    • A table summarizing genes, loci, and primer sequences, or a reference to where they are available.
    • A short justification for using Sanger sequencing rather than next-generation sequencing given the study period (2019–2021).
  • State the criteria used to classify isolates as “inferred mutations.” The text mentions WT/MUT absence but does not specify whether borderline bands or weak hybridization were re-tested.

Results

  • Several numerical inconsistencies require verification.
    • The percentages in Table 1 (e.g., 48.3%, 26.1%, 25.6%) do not always sum logically to 100%; please confirm denominators.
    • The text alternates between 41/623 = 6.6% and later 38/623 = 6.1% for resistant isolates. Clarify whether sequencing reclassification explains this difference.
  • Table 2 is valuable but dense. Consider reorganizing it by gene locus (gyrA, gyrB, rrs) and highlight mutation frequency rather than listing all background genotypes. This will improve readability.
  • The flowchart (Figure 1) lacks clear labels for “MTBDRplus” vs. “pDST.” Consider re-drawing it with a simpler vertical format to clarify the progression from total isolates to resistant subsets.
  • Demographic data (Table 3) are useful but would be strengthened by adding a comparison column showing the proportion of resistant vs. non-resistant patients if data are available.

Discussion

  • The discussion synthesizes findings well but should have stronger interpretation of public health implications, for example, what low prevalence of second-line resistance means for Brazil’s adoption of the BPaLM regimen.
  • The text repeats several descriptive results. Focus instead on the biological interpretation of gyrA D94G and S91P dominance and how these correlate with clinical outcomes.
  • The discussion would also improve with a brief comparison to other Latin American datasets (e.g., Peru, Colombia) rather than repeatedly citing the same São Paulo studies [4–5].
  • Add a concise paragraph on diagnostic performance, emphasizing the sensitivity/specificity or potential false-resistance rates of MTBDRsl relative to sequencing, since this is a key methodological strength of the study.
  • Consider consolidating the final three paragraphs (lines 280–296) into a formal “Study Limitations” subsection, covering incomplete MTBDRsl coverage, partial sequencing, and absence of phenotypic confirmation.

Conclusions

  • The conclusion is concise and appropriate, though slightly repetitive. Condense to a sharper, two-sentence takeaway emphasizing: 1 The need for routine second-line resistance testing among MDR-TB cases and 2) The critical role of confirmatory sequencing for inconclusive LPA results.
  • Include a short statement on future directions, such as expanding whole-genome sequencing or integrating data into the national surveillance network.

Figures and Tables

    • Ensure consistent use of decimal places and units (% vs. n/N) across tables.
    • Add titles above rather than below each table per MDPI style.
    • Replace the current Figure 1 placeholder with a properly labeled flowchart.
    • Consider a bar chart summarizing mutation frequencies (D94G, A90V, S91P) for rapid visual comparison.

English Language and Style

  • The English is clear overall. Change “referred to our laboratory by laboratories” → “submitted from regional laboratories”.
  • Consistency issues: alternate use of “gyrA D94G” and “D94G mutation in gyrA.” Choose one style throughout.
Comments on the Quality of English Language
  • The English is clear overall. Change “referred to our laboratory by laboratories” → “submitted from regional laboratories”.
  • Consistency issues: alternate use of “gyrA D94G” and “D94G mutation in gyrA.” Choose one style throughout.

Author Response

Reviewer 4

Comments and Suggestions for Authors

Introduction

  • The introduction provides adequate global and national context, but it should clearly articulate the knowledge gap, why São Paulo’s molecular resistance data remain underexplored and how this study adds to regional surveillance literature.

Response: We thank the reviewer for this valuable comment. Molecular tests for drug resistance were only introduced into the routine diagnostic workflow in São Paulo in 2019, and later in other Brazilian states. In addition, there are no national drug resistance surveys currently available in Brazil, which contributes to the limited knowledge regarding the molecular epidemiology of TB resistance. This study helps to fill this gap by providing regional data that can support surveillance efforts. All this information was included in the end of Introduction.

  • The WHO definitions of pre-XDR and XDR-TB are included, but the transition from global guidance to local relevance is abrupt. Strengthen by briefly connecting how these evolving definitions affect diagnostic policies in Brazil.

Response: The updated WHO definitions of pre-XDR and XDR-TB have impacted the diagnostic algorithm in Brazil. Under the previous definitions, routine diagnosis of these forms was already performed using either phenotypic DST or MTBDRsl assays. With the new definitions, MTBDRsl now detects only pre-XDR TB, and in São Paulo, phenotypic testing for bedaquiline and linezolid was implemented at the end of 2024 for detection of XDR-TB. Additionally, the BPaL regimen was introduced in Brazil in 2024, further emphasizing the need for accurate detection of resistance to guide treatment strategies. This information was included in the Introduction.

  • Citations [4]–[5] and [10] overlap conceptually with this study. You could emphasize what distinguishes your dataset (e.g., post-2021 MTBDRsl roll-out) to highlight originality.

Response: Changes were made to emphasize that our dataset is more recent than references [4 and 5].

  • Add a concise statement of hypothesis or primary objective at the end of the introduction (e.g., “We hypothesized that…”) to orient readers toward the analytic focus.

Response: The statement was included at the end of the Introduction.

Methods

  • The methods are well structured but require clearer justification for inclusion/exclusion of isolates. For example, specify the rationale for excluding duplicate isolates from the same patient and describe how discordant results were handled statistically or qualitatively.

Response: We thank the reviewer for this comment. To accurately estimate the prevalence of resistance, duplicate isolates from the same patient with identical mutation profiles were excluded from the analysis. Discordant results between MTBDRsl and gene sequencing were analyzed qualitatively, with each case reviewed to determine the likely cause of discrepancy, such as heteroresistance or limitations of the assay. No additional statistical adjustments were applied, as the focus was on descriptive characterization of resistance patterns. This information was included in the Methods section.

  • The sample size (n = 13,557) is impressive, yet the selection pathway (from initial isolates to analysed subset) should be summarized in a short paragraph before Figure 1 for transparency.

Response: This information was added before Figure 1.

  • The gene targets and sequencing protocols are described in technical detail, but the section should have:
    • A table summarizing genes, loci, and primer sequences, or a reference to where they are available.

Response: We appreciate the reviewer’s valuable suggestion.  In response, we have added the sequencing information for the gyrA, gyrB and rrs genes, along with the references describing the primer sequences, genes, and loci used in this study.

  • A short justification for using Sanger sequencing rather than next-generation sequencing given the study period (2019–2021).

Response: This information was added.

  • State the criteria used to classify isolates as “inferred mutations.” The text mentions WT/MUT absence but does not specify whether borderline bands or weak hybridization were re-tested.

Response: We thank the reviewer for this comment. Inconclusive LPA results were retested and only isolates with truly inferred mutations—based on the absence of wild-type and mutation probes—were submitted to Sanger sequencing. This information was included.

Results

  • Several numerical inconsistencies require verification.
    • The percentages in Table 1 (e.g., 48.3%, 26.1%, 25.6%) do not always sum logically to 100%; please confirm denominators.

Response: We have recalculated the percentages accordingly.

  • The text alternates between 41/623 = 6.6% and later 38/623 = 6.1% for resistant isolates. Clarify whether sequencing reclassification explains this difference.

Response: A sentence was added explaining the sequencing reclassified 3 isolates as wild-type.

  • Table 2 is valuable but dense. Consider reorganizing it by gene locus(gyrA, gyrB, rrs) and highlight mutation frequency rather than listing all background genotypes. This will improve readability.

Response: We thank the reviewer for the comment. Table 2 is already organized by gene locus (gyrA, gyrB, rrs). We have decided to retain the genotype information, as these data are important given the scarcity of molecular resistance information in Brazil.

  • The flowchart (Figure 1) lacks clear labels for “MTBDRplus” vs. “pDST.” Consider re-drawing it with a simpler vertical format to clarify the progression from total isolates to resistant subsets.

Response: We have redrawn the flowchart accordingly.

  • Demographic data (Table 3) are useful but would be strengthened by adding a comparison column showing the proportion of resistant vs. non-resistant patients if data are available.

Response: We thank the reviewer for the suggestion. Data on non-resistant patients are not available, and collecting such information for the large number of cases would require substantial time and resources. Therefore, we were unable to include a comparison column in Table 3.

Discussion

  • The discussion synthesizes findings well but should have stronger interpretation of public health implications, for example, what low prevalence of second-line resistance means for Brazil’s adoption of the BPaLM regimen.

Response: We appreciate the reviewer’s comment. We have expanded the discussion to highlight the public health implications of our findings. The low prevalence of second-line drug resistance in our setting supports the safe implementation of the BPaL regimen in Brazil, as well as other individualized regimens containing fluoroquinolones.

  • The text repeats several descriptive results. Focus instead on the biological interpretation of gyrA D94G and S91P dominance and how these correlate with clinical outcomes.

Response: We appreciate the reviewer’s suggestion. The revised Discussion has been streamlined to reduce descriptive repetition and to emphasize the biological interpretation of the most prevalent mutations. We now highlight that gyrA D94G is associated with high-level fluoroquinolone resistance and has been linked to poorer clinical outcomes, whereas S91P generally confers lower or intermediate resistance, potentially allowing partial response to moxifloxacin. These differences were incorporated to better contextualize the clinical relevance of the detected mutations.

  • The discussion would also improve with a brief comparison to other Latin American datasets (e.g., Peru, Colombia) rather than repeatedly citing the same São Paulo studies [4–5].

Response: We appreciate the suggestion. A brief comparison with data from other Latin American countries has been added to the Discussion.

  • Add a concise paragraph on diagnostic performance, emphasizing the sensitivity/specificity or potential false-resistance rates of MTBDRsl relative to sequencing, since this is a key methodological strength of the study.

Response: We have rewritten the paragraph on MTBDRsl false resistance in comparison to sequencing, so we hope it is clearer now.

  • Consider consolidating the final three paragraphs (lines 280–296) into a formal “Study Limitations” subsection, covering incomplete MTBDRsl coverage, partial sequencing, and absence of phenotypic confirmation.

Response: We have consolidated the final three paragraphs into a dedicated “Study Limitations” subsection. This section now explicitly addresses the incomplete MTBDRsl coverage for some isolates, the partial sequencing of gene targets, and the lack of phenotypic drug susceptibility testing for second-line drugs.

Conclusions

  • The conclusion is concise and appropriate, though slightly repetitive. Condense to a sharper, two-sentence takeaway emphasizing: 1 The need for routine second-line resistance testingamong MDR-TB cases and 2) The critical role of confirmatory sequencing for inconclusive LPA results.

Response: As suggested, we have revised the conclusion to make it more concise and focused, emphasizing the need for routine second-line resistance testing among MDR-TB cases and the importance of confirmatory sequencing for inconclusive LPA results.

  • Include a short statement on future directions, such as expanding whole-genome sequencing or integrating data into the national surveillance network.

Response: We have added a statement on future directions, highlighting the importance of expanding next generation sequencing implementation and integrating molecular resistance data into the national TB surveillance network to improve monitoring and guide treatment strategies.

Figures and Tables

  • Ensure consistent use of decimal places and units (% vs. n/N) across tables.

Response: We have revised all tables to ensure consistency in the use of decimal places and in the presentation of values, standardizing percentages (%) and proportions (n/N) throughout the manuscript.

  • Add titles above rather than below each table per MDPI style.

Response: We have reformatted all tables to place the titles above, in accordance with the MDPI style guidelines.

  • Replace the current Figure 1 placeholder with a properly labeled flowchart.

Response: We have replaced the current Figure 1 placeholder with a fully labeled flowchart that clearly represents the study workflow.

  • Consider a bar chart summarizing mutation frequencies (D94G, A90V, S91P) for rapid visual comparison.

Response: We have added a bar chart with gyrA mutations frequency, as recommended.

English Language and Style

  • The English is clear overall. Change “referred to our laboratory by laboratories” → “submitted from regional laboratories”.

Response: Change was made.

  • Consistency issues: alternate use of “gyrA D94G” and “D94G mutation in gyrA.” Choose one style throughout.

Response: We kept “gyrA D94G” throughout the manuscript.

Comments on the Quality of English Language

  • The English is clear overall. Change “referred to our laboratory by laboratories” → “submitted from regional laboratories”.

Response: Change was made.

  • Consistency issues: alternate use of “gyrA D94G” and “D94G mutation in gyrA.” Choose one style throughout.

Response: We kept “gyrA D94G” throughout the manuscript.
